# In Depth Topological Analysis of Arabidopsis Mid-SUN Proteins and Their Interaction with the Membrane-Bound Transcription Factor MaMYB

**DOI:** 10.3390/plants12091787

**Published:** 2023-04-27

**Authors:** Bisa Andov, Aurelia Boulaflous-Stevens, Charlotte Pain, Sarah Mermet, Maxime Voisin, Camille Charrondiere, Emmanuel Vanrobays, Sylvie Tutois, David E. Evans, Verena Kriechbaumer, Christophe Tatout, Katja Graumann

**Affiliations:** 1Department of Biological and Molecular Sciences, Faculty of Health and Life Sciences, Oxford Brookes University, Oxford OX3 0BP, UK; 2CNRS, Inserm, GReD Clermont-Ferrand, Université Clermont Auvergne, 63001 Clermont-Ferrand, France

**Keywords:** mid-SUN protein, nuclear envelope, endoplasmic reticulum, higher plant, Arabidopsis, maMYB

## Abstract

Mid-SUN proteins are a neglected family of conserved type III membrane proteins of ancient origin with representatives in plants, animals, and fungi. Previous higher plant studies have associated them with functions at the nuclear envelope and the endoplasmic reticulum (ER). In this study, high-resolution confocal light microscopy is used to explore the localisation of SUN3 and SUN4 in the perinuclear region, to explore topology, and to study the role of mid-SUNs on endoplasmic reticulum morphology. The role of SUN3 in the ER is reinforced by the identification of a protein interaction between SUN3 and the ER membrane-bound transcription factor maMYB. The results highlight the importance of mid-SUNs as functional components of the ER and outer nuclear membrane.

## 1. Introduction

Sad1/Unc84 (SUN) domain proteins are conserved across all eukaryotes and can be classified into two subgroups, Cter-SUN proteins, with a C-terminal SUN domain, and mid-SUNs, in which the SUN domain occupies a central location in the protein [1,2]. Cter-SUNs are inner nuclear membrane (INM)-localised and interact with outer nuclear membrane (ONM)-localised plant Klarsicht/Anc1/Syne homology (KASH) domain proteins to form Linker of Nucleoskeleton and Cytokseleton (LINC) complexes with multiple functions [3,4,5,6,7,8,9]. On the other hand, mid-SUN proteins are type III transmembrane proteins and information on their functions is limited to a small number of systems [2,10,11,12,13,14,15,16,17,18,19]. In yeast and metazoans, they are predominantly ER-located with potential functions in integral membrane protein folding in the ER [12]. In maize, three mid-SUN proteins, ZmSUN3-5, were identified, two (ZmSUN3 & 4) being expressed ubiquitously while ZmSUN5 is most abundant in pollen [2]. This expression pattern was also found to be true for the three Arabidopsis mid-SUNs (SUN3, SUN4, and SUN5) [13] and for GbSUN5 in cotton [20]. The Arabidopsis mid-SUNs have been shown to interact with LINC components–Cter-SUNs and KASH proteins–and to play a role in nuclear morphology and shape [13]. Arabidopsis SUN3 and SUN4 were found to localise in both the NE and ER, which was also shown for chickpea mid-SUN protein CaSUN1 [14]. CaSUN1 was identified in a screen investigating potential components of the dehydration response pathway, which involves the ER-localised unfolded protein response pathway [14]. Recently, a triple *sun3/sun4/sun5* mutant was shown to have impaired pollen tubes and plant growth with leucine-rich receptor kinases normally destined for the plasma membrane missorted to acidic compartments [18]. Mid-SUNs were identified as part of a POD1–SUN–CRT3 chaperone complex guarding the ER sorting of the LRR receptor. Xue et al. [17] found that loss of SUN3/4/5 caused tubule-to-sheet transition and overexpression that led to ER deformation, suggesting that the coiled-coil domain contributes to the maintenance of ER morphology.

Taken together, the previous work described above leaves a number of areas of uncertainty about the location, topology, and function of mid-SUN proteins in plants. A NE localisation has been suggested, with evidence of interaction with NE components, while there is mounting evidence for roles in the endoplasmic reticulum, including as part of a chaperone complex involved in the ER sorting of a plasma membrane receptor and in the determination of the morphology of the ER. Here, we explore in detail the location of mid-SUN proteins at the outer nuclear membrane and ER. We explore the role of mid-SUNs in determining the structure and properties of the ER and provide novel insights into the topology of mid-SUN protein domains. We finally provide evidence demonstrating that mid-SUN proteins interact with an ER-bound transcription factor, maMYB.

## 2. Results

### 2.1. Mid-SUN Proteins Are Enriched in the ONM and ER

With previous reports of ER and NE localisation of plant, yeast, and metazoan mid-SUN proteins, we firstly aimed to better understand the subcellular localisation of Arabidopsis SUN3 and SUN4. We therefore compared their ER/NE ratio using fluorescence intensity (FI) of ER and NE markers in transiently expressing *Nicotiana benthamiana* (*N. benthamiana*) leaf epidermal cells by establishing the intensity of fluorescence in regions of interest (ROI) in the NE and cortical ER. To achieve this, AtSUN2 and RETICULON1 (RTN1) were used as NE and ER markers, respectively. SUN2-YFP was predominantly localised to the NE (NE/ER ratio = 3.3 ± 0.28) while the ER marker GFP-RTN1 was predominantly ER-localised (1.2 ± 0.09) as expected. RFP-SUN3 (1.4 ± 0.09) and SUN4-RFP (1.2 ± 0.07) were also found to be significantly enriched in the ER with values very similar to RTN1 (Figure 1A). To dissect the subcellular localisation of the two mid-SUN proteins more precisely, we resolved their membrane enrichment by determining fluorescence intensity (FI) on line profiles. Line profiles as illustrated in Figure 1 and Appendix A were defined using calnexin-mCherry (CXN-mCherry) as a marker of the ER [21]. In this case, SUN2 fluorescence was significantly nearer the centre of the nucleus (0.067 ± 0.012 µm) in good agreement with its predominant INM localization, while SUN3 peak fluorescence was external to that of CXN-mCherry (GFP-SUN3: −0.036 ± 0.007 µm and SUN3-GFP: −0.028 ± 0.007 µm) with SUN4 colocalised with it (GFP-SUN4: 0 ± 0.009 µm) suggesting strongly that the mid-SUNs are predominantly localised at the ONM/ER (Figure 1B).

### 2.2. Quantifying the Effect of SUN3 and 4 OverExpression on ER Architecture

SUN3 and 4 have been implicated in maintaining ER morphology [18]. To further explore the potential function of SUN3 and 4 as ER morphogens, we compared ER morphology of control cells expressing GFP-HDEL alone (Figure 2A) with N- and C-terminally tagged SUN3 and SUN4 (Figure 2B–E) using the ER structure quantification software AnalyzER [22]. An initial analysis by MANOVA revealed significant changes in ER structure on overexpression of both SUN proteins (Pillias trace, *p*-value = 9.31 × 10^−12^ and Roy’s largest root, *p*-value = 1.90 × 10^−22^). Plotting the first two discriminate canonical variables (Figure 2F) and a dendrogram plot of group means (Figure 2G) indicated that SUN3 and SUN4 overexpression results in different modifications to ER structure. The changes induced by overexpression were significant but not large in magnitude and many of the results were within the 95% confidence limit of the control group (Figure 2F). Which terminal was tagged appeared to have limited impact on the effect of SUN3 and 4 on ER structure. ANOVA comparisons, with an applied Bonferroni correction for multiple comparisons, revealed changes in tubule length, cisternae area and shape, and polygonal region size and shape (Figure 2H). Interestingly, SUN3 and 4 appeared to have largely opposing impacts on ER structure, with SUN3 overexpression significantly increasing mean cisternae area (by at least 85.3%) whilst SUN4 overexpression reduces the mean cisternal area by at least 21.15%, though this change is not statistically significant (Appendix A).

SUN4 overexpression also caused GFP-HDEL to localise preferentially to small pockets spread throughout the ER. This phenotype is similar to what has been observed previously in overexpression of membrane constricting members of the reticulon protein family [23]. Using high-resolution confocal microscopy imaging, we were able to observe that whilst CXN-mCherry, which normally localises to ER and ONM membranes, has an even distribution of lumenal contents along its tubules (visualised by GFP-HDEL). This was not the case for RFP-SUN4, SUN4-RFP, and RFP-RTN1 (Figure 3A–D). Tracing the intensity of the luminal marker GFP-HDEL along tubules also containing RFP-SUN4 and SUN4-RFP showed inconsistent luminal contents, with patches of more or less GFP-HDEL visible. However, when compared to RFP-RTN1 overexpression, there were no long stretches of tubule completely lacking GFP-HDEL. This suggests that membrane-bound SUN4 was able to induce some ER lumen constriction resulting in GFP-HDEL patches—but not to the same extent and potential mechanism as RFP-RTN1 (Figure 3E). In addition, peaks in CXN-mCherry and GFP-HDEL show some co-occurrence along the length of tubules. The same is true of SUN4 and GFP-HDEL, whilst RFP-RTN1 and GFP-HDEL have peaks in intensity that do not co-occur, as GFP-HDEL is forced into areas with minimal RFP-RTN1 expression (Figure 3F). This suggests that areas of high SUN4 in the ER membrane do not correlate well with areas of constriction, as they do on RFP-RTN1 overexpression.

### 2.3. Using roGFP to Determine the Membrane Topology of SUN3 and 4

Mid-SUN proteins contain three transmembrane domains, coiled-coil regions, and a SUN domain with the CC and SUN domains involved in protein–protein interaction [13]. The membrane topology of both SUN3 and SUN4 proteins was examined using the redox sensitive fluorescent tag roGFP2 (hereafter referred to as roGFP; [24]). roGFP has two excitation peaks at 405 nm and 488 nm [24], and the efficiency of each excitation peak is dependent on the redox state of the surrounding environment. In the more oxidised environment of the ER lumen, cysteine residues in roGFP form disulphide bonds, improving excitation at 405 nm [25,26,27]. In the more reduced environment of the cytosol, excitation is most efficient at 488 nm [26,27]. Similar to previous studies, cytosolic roGFP transiently expressed in *N. benthamiana* showed almost no excitation at 405 nm and significant excitation at 488 nm (Figure 4A), a ratio of 0.1 ± 0.009 (Figure 4K). When roGFP was localised to the lumen of the ER, using the ER retention signal HDEL, approximately equivalent excitation at both 405 nm and 488 nm (Figure 4B), a ratio of 1.2 ± 0.03 (Figure 4K), was observed. The roGFP-SUN3 (Figure 4C) had a similar excitation profile to cytosolic roGFP, ratio 0.2 ± 0.03, while SUN3-roGFP (Figure 4D), had a similar excitation profile to ER lumenal roGFP (ratio 8.9 ± 0.08). SUN4 followed the same patterns, with roGFP-SUN4 showing similar excitation to cytosolic roGFP (ratio 0.1 ± 0.008; Figure 4E), and SUN4-roGFP showing excitation similar to the ER lumen (ratio 1.1 ± 0.08; Figure 4F). These results indicate that the N-terminal tails of SUN3 and SUN4 are located in the cytosol, the SUN domain and coiled-coil domain are present in the ER lumen, followed by two transmembrane domains, leaving the C-terminal tail of SUN3 and SUN4 located in the ER lumen (Figure 4G).

This topology analysis implies that the SUN domain of SUN3 and SUN4 is not accessible to cytosolic and soluble nuclear proteins, whereas it can interact with proteins located in the ER lumen and nuclear periplasm via its SUN and coiled-coil domain and more limited in the cytoplasm via its N-terminus and a cytosolic loop.

### 2.4. Arabidopsis Mid-SUN3 Interacts with the ER-Localised Membrane Transcription Factor maMYB

To gain further insights into the possible function of SUN3 at the ER, a Membrane Yeast Two-Hybrid (MYTH) screen was performed to search for SUN3 interactors (Appendix A). A transcription factor localized at the ER, maMYB [28], was detected. maMYB contains two transmembrane domains at the N-terminus and two SANT domains at the C-terminus [28]. maMYB had the strongest and most reproducible interaction with SUN3. Weaker interactions were detected between maMYB and SUN1, as well as with SUN2 and SUN5, but no interaction was recorded with SUN4 (Figure 5A) suggesting a specific role for the interaction. Interactions were then confirmed in planta using apFRET. Transiently expressed fluorescent protein fusions of SUN proteins and maMYB were colocalised in the ER and NE. Protein–protein interactions were determined by calculating the FRET efficiency [E_F_] between each pair of proteins measured in ROI selected at the ER and the NE. Protein interactions were detected between CFP–maMYB or maMYB–CFP and YFP–SUN3 while no interaction was detected between maMYB and SUN1, SUN2, or SUN4 in planta (Appendix A). Despite co-localisation at the NE and ER, only SUN3 interacted with maMYB in planta whilst SUN4 did not, in agreement with the MYTH results. Deletion analysis performed in MYTH on maMYB and SUN3 suggested the interaction requires the N-terminal domains of both SUN3 and maMYB (Figure 5B,C).

To confirm the relevance of the interaction between maMYB and SUN3, co-occurrence of the two proteins was searched across a set of species chosen to represent the green lineage (Appendix A). According to the Orthologous Matrix (OMA) and EggNOG databases [29,30], SUN3 and maMYB have a large number of orthologues across plant species, although the mid-SUN domain was already present in unicellular green algae while maMYB appears later in mosses (Appendix A). Interestingly, the transmembrane domains of maMYB and SUN3 as predicted by TMHMM-2.0 [31] are well conserved in all orthologues. Altogether, the localization and interaction between maMYB and SUN3 and the conservation of transmembrane domains during evolution suggest that the two proteins may have an important function in the ER and potentially the outer nuclear membrane.

### 2.5. Disruption of SUN3 and maMYB Affects Root Hair Elongation

To understand whether SUN3 and maMYB could act in the same pathway, we studied the genetic interaction between *SUN3* and *maMYB*. To this end, a *mamyb-1* mutant was generated using CRISPR/Cas9 [32] by targeting the 5′ *maMYB* coding sequence. Transcripts from the *mamyb-1* mutant allele are produced but given the stop codon occurring after 26 amino acids, no maMYB protein was revealed by Western Blot suggesting that it is a true null mutation (Appendix A). Homozygous *mamyb-1* plants were crossed with the *sun3-1* mutant [13] to obtain the double mutant, *sun3-1 mamyb-1.* Both single and double mutants did not show any obvious growth defects. Disruption of *maMYB* was previously shown to affect root hair length but not root hair initiation using RNAi lines [28]. As previously described, we did not observe significant differences in root hair numbers among the tested genotypes. Therefore, root hair elongation was investigated using a similar experimental procedure as described by Slabaugh et al. [28] (Figure 6A). In this assay, each mutant line was normalised with the wild-type Col0 line (mutant/Col0 ratio set as 1) and root hair length expressed as a ratio between a given mutant and Col0 plants grown simultaneously. Root hair length was reduced in the *sun3-1* (64.46 ± 0.26%), *mamyb-1* (49.38 ± 0.25%), and in *sun3-1 mamyb-1* (53.11 ± 0.21%) lines while overexpression of maMYB-1-YFP under a 35S promoter had no significant impact on root hair length (89.44 ± 0.34%) (Figure 6B). These results suggested that SUN3 and maMYB are acting in the same pathway to promote root hair elongation as no additive effect was observed in double mutants versus single mutants.

## 3. Discussion

Knowledge of the localisation and function of mid-SUN proteins is expanding rapidly, with growing evidence from numerous studies, especially of higher plant systems. Here, we showed that mid-SUN proteins are functional constituents of the ER and ONM with luminal SUN and coiled-coil domains, as well as functions in ER shaping and specialised transcription factor binding.

Using fluorescence intensity of ER and NE markers, we showed the enrichment of SUN3 and SUN4 in the ONM and in the ER, but not in the INM (Figure 1). As the ONM is continuous with the ER through junctional regions, this is not unexpected and other ER markers (such as Calnexin) are found in both membranes. However, mid-SUNs appear to have specific functionality in both membranes as they previously have been shown to interact with (ONM)-localised KASH proteins and Cter-SUNs [13]. In yeast, the sole mid-SUN protein, SLP1, is part of a complex with the YERP65 protein recruiting the Cter-SUN protein Mps3 to the NE [12]. Whether plant mid-SUN proteins have a similar chaperoning function in transporting NE components to their destined membrane is currently unknown but an enticing concept to investigate.

While both are localised in the ER and ONM, SUN3, but not SUN4, was previously shown to have different mobile behaviours in the ER and NE indicating different protein interactions in these membranes [13]. This mirrors our findings that SUN3 but not SUN4 is able to interact with maMYB (Figure 5) providing an example of such differing interactions.

The transcription factor maMYB is a member of the R2R3-MYB subgroup of the homeodomain-like MYB superfamily. This gene family is a large, plant-specific subfamily of the MYB gene family in eukaryotes that all have DNA binding functionality in the MYB domain [33]. Arabidopsis maMYB is a membrane-bound transcription factor involved in plant development and responses to environmental stress, including cell division [34], hormone signalling [35], and root hair elongation [28]. Here, we found that maMyb and SUN3 interact both in vitro and in planta in the ER (Figure 5 and Appendix A). Evolutionary analysis of this SUN3–maMYB complex shows the presence of SUN3 through the green lineage while maMYB emerges in the land plants (Appendix A), reinforcing the idea of a fundamental function in eukaryotic plant cells for mid-SUN proteins and a later specialised function for maMYB. This specialisation is further suggested by the fact maMYB was only shown to interact with SUN3 and not SUN4 (Figure 5 and Appendix A). To investigate whether there is a functional as well as evolutionary link between the two, we examined root hair length. It is known that silencing of maMYB using the RNAi approach results in a significant reduction of root hair length in *Arabidopsis* but similar root hair density compared to wild-type [28]. We were able to replicate this by showing that *mamyb-1* had a severely reduced root hair length (Figure 6). We also observed a similar significant reduction in root hair length in the *sun3-1* mutant compared to Col0 (Figure 6). The simultaneous inactivation of *SUN3* and *maMYB* in the double mutant *sun3-1 mamyb-1* resulted in slightly shorter root hairs compared to the single *sun3-1* mutant indicating that SUN3 and MaMYB are both involved in root hair elongation. Reduced root hair length was previously observed in a triple *sun3 sun4 sun5* Arabidopsis mutant, in which the phenotype could be rescued by expression of SUN4-GFP [18]. This redundancy was not observed in our study with the shorter root hair length phenotype resulting from the absence of SUN3 only.

Protein interactions are mediated by domains and motifs and so to understand better how SUN3 and SUN4 are functional in the ER and ONM, it is essential to look at their topology. The roGFP assays in this work (Figure 4) as well as previous BiFC analysis [18] suggest that mid-SUN proteins are oriented with the SUN domain, coiled-coil domain, and C-terminus in the ER and NE lumen, and with the N-terminus in the cytosol (Figure 4) with a potential for the protein to bridge opposite ER membranes. This topology analysis implies that the SUN domain of SUN3 and SUN4 is not accessible to cytosolic and soluble nuclear proteins, whereas it can interact with proteins located in the ER and NE lumen. This lends support to previous observations showing mid-SUNs to interact with KASH proteins where the luminal KASH and SUN proteins mediate this interaction [13]. In addition, it was shown that the luminal coiled-coil domain of SUN2 is required for SUN2–SUN3 interactions. It is therefore likely that the SUN2–SUN3 interactions also occur in the NE lumen mediated by their respective coiled coils.

While the bulk of the mid-SUN protein is localised in the ER/NE lumen, protein interactions are also possible on the cytoplasmic side and via the TMDs in the membrane. Indeed, protein domain deletion analysis using MYTH suggested the interaction requires the N-terminal domains of both SUN3 and maMYB (Figure 5). As the N-terminal domains of both interactors used in the study are composed of a cytosolic N-terminal tail and transmembrane domains, the interaction can take place either in the cytosol and/or in the ER membrane. The MYB domain, located toward the cytosol, does not interact with SUN3, and remains free for a proteolytic activation and transport to the nucleus.

Their functional presence in the ER indicates that mid-SUN proteins might have an effect on ER structure. Indeed, previously Xue et al. [17] showed that loss of SUN3, SUN4, and SUN5 leads to increased cisternal area, and that overexpression of SUN4 results in the formation of enlarged isolated bubbles of lumenal contents. To explore their role as potential ER morphogens, we used AnalyzER to quantify effects on ER structure. Our results demonstrated that SUN3 and SUN4 have different, opposing effects on ER structure, with SUN3 inducing an increase in cisternal area and SUN4 resulting in an abundance of tubules with a partial constriction phenotype (Figure 2). Climp63, a mammalian lumenal spacer, has the effect of inducing cisternae proliferation on overexpression and acts to stabilise across larger cisternae [36]. This is similar to the impact of SUN3 on ER structure and of potential interest as no plant Climp63 has been reported so far. However, mammalian Climp63 localises preferentially to ER cisternae, as tubule formation by membrane bending is reliant on the reticulon proteins that shape the ER through positive membrane curvature.

The partial constriction phenotype of SUN4 is particularly interesting. It was previously shown that a loss of *sun3 sun4 sun5* could not be compensated for by RTN overexpression [18] suggesting that the function of the mid-SUN proteins in membrane shaping is different to known ER morphogens. Our results further support this by showing that SUN4 overexpression, while appearing to cause a mild constriction phenotype, does not act in a similar manner to RTN1 (Figure 3). The magnitude of the modifications to ER structure also suggest that although SUN3 and 4 act to shape the ER, their impact when compared to known plant ER morphogens such as the Lunapark proteins [37], the reticulon protein family [23], and ROOT HAIR DEFECTIVE 3 (RHD3) [38] is relatively low. However, this more subtle effect on membrane morphology paired with their functional presence at the NE may indicate that the mid-SUNs could have potential functions in NE morphology and dynamics.

Taken together, this work suggests important roles for mid-SUN proteins in the ER and ONM. Studies to date, including this, reveal interactions with Cter-SUNs and KASH proteins (potentially at the ONM [13]), a subtle but distinctive role in ER morphology, interactions with a membrane transcription factor, the unfolded protein response pathway (Jaiswal et al., 2014) in pollen tube and root hair growth, and a chaperone complex guarding the ER sorting of LRR receptor [17]. Given their primitive evolutionary origins, it is a diverse set of functions and it will be interesting to unravel the molecular processes underlying them, especially for the roles of plant mid-SUN proteins.

## 4. Materials and Methods

### 4.1. Plant Material and Growth Conditions

*Arabidopsis thaliana* Col0 plants were grown from seeds surface sterilized using 0.01% SDS and 70% ethanol for 5 min, and washed twice with pure ethanol, then dried on Whatman paper. Dried seeds were transferred to half strength Murashige and Skoog media including vitamins, and supplemented with 1% sucrose, 1% Phytagel, and adjusted at pH 5.7, then stratified for 3 days at 4 °C in the dark. Arabidopsis seedlings were grown at 23 °C, 16 h photoperiod under illumination of 100 μE. *N. benthamiana* plants were grown on soil in a greenhouse in the same conditions.

### 4.2. Accessions Numbers, T-DNA Mutants and 35S::MaMYB-vYFP Line

Accession numbers for this study include *SUN1* (AT5G04990), *SUN2* (AT3G10730), *SUN3* (AT1G22882), *SUN4* (AT1G71360), *SUN5* (AT4G24950), and *maMYB* (AT5G45420). The *sun3* insertion T-DNA line Flag_374A03 (*sun3-1*) from the Salk collection was obtained from the European *Arabidopsis* Stock Centre. As Flag_374A03 line was in the *Ws* (*Wassilewskija*) background [13], three successive generations of outcross with wild-type Col0 plants were used to generate a near isogenic Flag_374A03 Col0 line. The T-DNA insertion was genotyped by PCR. A line overexpressing maMYB fused upstream to venus YFP under the 35S cauliflower promoter was provided by Prof F Brandizzi (maMYB-vYFP) [28].

### 4.3. Molecular Cloning and CRISPR/Cas9-Mediated Genome Editing

Arabidopsis *SUN3* and *SUN4* cDNA were cloned into the N-terminal roGFP2 (hereafter referred to as roGFP) vector pCM01 and C-terminal roGFP vector pSS01 created by Brach et al. (2009) using Gateway cloning technology (Invitrogen, Waltham, MA, USA).

For fluorescent fusions *SUN3* and *SUN4* cDNA was cloned into the vectors pB7RWG2 (C-terminal mRFP), pB7WGR2 (N-terminal mRFP), pB7FWG2 (C-terminal eGFP), pB7WGF2 (N-terminal eGFP) [39].

The *mamyb-1* mutant was generated using the CRISPR/Cas9 system [32]. A 20-nucleotide guide targeting the 5′ of the *mamyb* coding sequence (5′-GCAGAAGATGAACTCCGTTC-3′) was selected using Crispr-Plant (http://omap.org/crispr2/) [40] and cloned into pDe-Cas9 by Gateway LR clonase (Thermo Fisher Scientific, Waltham, MA, USA). The new binary vector was transformed into *Agrobacterium tumefaciens* (GV3101) and then into *A. thaliana* Col0 by the floral-dip method [41]. The selected knockdown *mamyb-1* mutant line results from an 8 nucleotides deletion leading to a premature stop codon and a truncated protein of 26 amino acids (Figure 1). This CRISPR-Cas9 generated mutant was crossed with the Flag_374A03 Col0 line to generate the *sun3-1 mamyb-1* double mutant. The CRISPr/Cas9 mutants are genotyped by PCR amplification and sequencing.

### 4.4. Subcellular Localisation of SUN3 and 4 Analysed via Proximity Distance

The subcellular localisation of SUN3 and SUN4 was measured using radial line profiles and confocal microscopy. Briefly, z-stacks of nuclei were collected and the medial point of each nucleus was selected for analysis, to limit the impact of membrane curvature on the analysis. Using the Radial Profile Extended plugin on ImageJ, ROIs of 30° for each nucleus were drawn and the fluorescence intensity (FI) was measured along each radial line profile. FI values of all the lines were averaged to produce an average FI plotted as distances (expressed in µm) along the radius. The mean distance for the peak intensities of SUN2-CFP, GFP-SUN3, SUN3-GFP, and GFP-SUN4 was then subtracted with the peak intensity of the ONM/ER marker calnexin-(CXN) mCherry to give proximity distances (in µm). Negative distance referred to markers enriched in ER while positive values are enriched in the nucleus. Significant differences between group means were determined first by a Kruskal–Wallis H test followed by Dunn’s test with Bonferroni correction as a post-hoc test, χ^2^(6) = 36.078, *p* = 2.662 × 10^−6^.

### 4.5. Quantifying Changes in ER Structure

*N. benthamiana* was infiltrated as described by [42]. Briefly, transformed Agrobacteria (GV3101) were grown overnight at room temperature. Cultures were pelleted by centrifugation at 2200× *g* for 3 min at room temperature, resuspended in infiltration buffer (50 mM MES, 2 mM Na_3_PO_4_.12H_2_O, 5mg mL^−1^ glucose, and 0.1 mM acetosyringone) and diluted to the appropriate optical density (OD_600_ 0.15 for SUN3 and 4 roGFP constructs; OD_600_ 0.1 for roGFP-HDEL and roGFP cytosolic constructs). All constructs were infiltrated alongside the silencing inhibitor P19 (OD_600_ = 0.1) (Lakatos et al., 2004) to improve expression. After infiltration by injection, plants were transferred to an incubator and maintained at 23 °C for 72 h before imaging.

ER structure analysis was performed on images collected using confocal microscopy on *N. benthamiana* transiently expressing fluorescently tagged proteins of interest. Analysis of ER structure was performed using the AnalyzER software, as described in [22]. In brief, confocal images were collected and imported into AnalyzER. Imaging conditions, including pixel size and channel order, were input into the software, together with estimated minimum and maximum tubule width. Images were upsampled to increase pixel density and to prevent pixelation error later in the analysis. A region of background outside the cell was measured, with the intensity of this region subtracted from the overall images. Image intensity was then enhanced by deploying a guided filter, to increase the intensity of the ER network. The area of analysis was masked out using thresholding, and within this region cisternae were identified using an opening function, followed by active contouring, to map the outline of cisternae onto the underlying image intensity. Any cisternae with an area of less than 0.3 µm^2^ were reclassified as puncta as defined by [43]. The tubular network was then enhanced using an edge preserving phase congruency filter [44]. The width of each tubule was calculated using integrated intensity granulometry. Multiple structure metrics were then calculated for each specific subregion of the ER, including: tubule length, the length of each tubule between junction points and cisternae in the ER; cisternae area, the area of each cisternae; polygonal region area, the area of each polygonal region within the ER, where the tubules have been thinned to a single pixel wide skeleton; elongation, the ratio of the major and minor axes; circularity, the ratio of the radius measured for the area of the region to the radius measured from the perimeter of the region and roughness, the ratio of the regions perimeter squared, to the area.

### 4.6. Tubule Morphology Quantification

Tubule morphology was calculated by measuring the diameter of tubules along their length, with the position determined by the tubular skeleton. The intensity of pixels along the tubular skeleton was measured and peaks in intensity identified. Troughs were identified using the same method, using an inverted intensity profile. Once peaks and troughs had been identified, the width of the tubule at that point was measured and the distance between peaks calculated. Using this data, it is possible to calculate the following measurements: peak/trough width, the estimated width of each peak/trough; peak/trough intensity, the mean intensity at the peak/trough; peak/trough covariance, the mean covariance of intensities of both channels, measured at the peak/trough intensity of each tubule. Positive values are an indicator of similar behaviour, whilst negative values suggest that the tubule intensities move in opposite directions.

### 4.7. Determination of SUN3 and SUN4 Topology Using roGFP

Samples of the leaf epidermis were imaged using a Zeiss PlanApo x100/1.46 NA oil immersion objective on a Zeiss 880 confocal microscope with Airyscan detector. Samples were excited using 405 and 488 nm lasers with line switching. The detection wavelength was set to between 500–588 nm. For improved signal to noise ratio, images were collected using 4× averaging. The mean intensity of an image was quantified by measuring the intensity in an ROI on the ER (or cytoplasm in the case of the cytoplasmic control) using FIJI. A region of background intensity was also measured in the area surrounding the ER, or outside the cell when using cytoplasmic controls. The background intensity was subtracted from the measured fluorophore intensity in both channels. Then, the intensity of the 405 nm image was divided by the intensity of the 488 nm image.

### 4.8. Membrane Yeast Two-Hybrid (MYTH)

Protein interaction of membrane proteins SUN3 and maMYB were performed using Split-Ubiquitin-based Membrane Yeast Two-Hybrid (MYTH) system as previously described [45]. Yeast media used were a standard Yeast Nitrogen Base (YNB) supplemented with amino acids and bases as required. Interactions were tested on Permissive Medium (PM: YNB without Leu and Trp) and Test Medium (TM: YNB without Leu, Trp, Ade, and His) using the NMY51 strain (*MATa*, *his3Δ200*, *trp1-901*, *leu2-3,112*, *ade2*, *LYS2::(lexAop)4-HIS3*, *ura3::(lexAop)8-lacZ*, *ade2::(lexAop)8-ADE2*, *GAL4*) and the following prey plasmid pPR3N (*2μ*, *TRP1*, *AmpR*), and bait plasmid pBT3N (*CEN*, *LEU2*, *KanR*). Full-length SUN constructs were as in [13]. *SUN* and *MaMYB* deletion constructs used for the MYTH were generated by ‘gap-repair’ homologous recombination in yeast [46] from cDNAs amplified using chimeric primer pairs containing 5′ ends with 35 bp of homology to the linearized yeast target plasmid. After digestion by SfI1, bait and prey plasmids and PCR-amplified cDNAs were cotransformed in yeast with a vector/insert of 1/3. All constructs were checked and sequenced. Screen for new SUN3 interacting proteins was performed using a cDNA library (DUALSYSTEM Biotech) cloned into the prey vector pDSL-Nx (*2μ*, *TRP1*, *AmpR*) as described in [13].

### 4.9. Conservation of MaMYB and mid-SUN3 and Phylogenetic Reconstruction

maMYB and SUN3 protein orthologues were selected in 16 species from green algae, basal land plants and angiosperms: *Arabidopsis thaliana*, *Arabidopsis lyrata*, *Brassica oleracea*, *Populus trichocarpa*, *Theobroma cacao*, *Prunus persica*, *Glycine max*, *Oryza sativa* sub sp *indica*, *Musa acuminata*, *Zea mays*, *Amborella trichopoda*, *Picea glauca*, *Selaginella moelledorffii*, *Physcomitrella patens*, *Chlamydomonas reinhardtii*, *Ostreococcus lucimarinus*. Orthologues were collected using OMA and EggNOG databases [29,30]. Graphic representation was performed using ITOL [47]. Signal peptide were predicted by SignalP-5.0 [48] and Phobius [49], transmembrane domain by TMHMM-2.0 [32]. The accession numbers of all orthologues are provided in Appendix A.

### 4.10. Root Hair Measurements

*Arabidopsis thaliana* wild-type (Col0), 3 mutant lines (*sun3-1*, *mamyb-1*, *sun3-1 mamyb-1*) and a *35S::maMYB-YFP* line were used for the root hair analysis. The 11-day-old seedlings, grown upright, were imaged using an Olympus XC30 digital camera mounted on an Olympus SZX12 stereomicroscope. Digital images were analysed with ImageJ software. Root hair measurements were taken according to Slabaugh et al. [28]. Root hairs were measured within the differentiation zone approximately 5 mm from the root tip and for each plant the five longest root hairs in focus were measured in five independent replicates.

## Figures and Tables

**Figure 1 plants-12-01787-f001:**
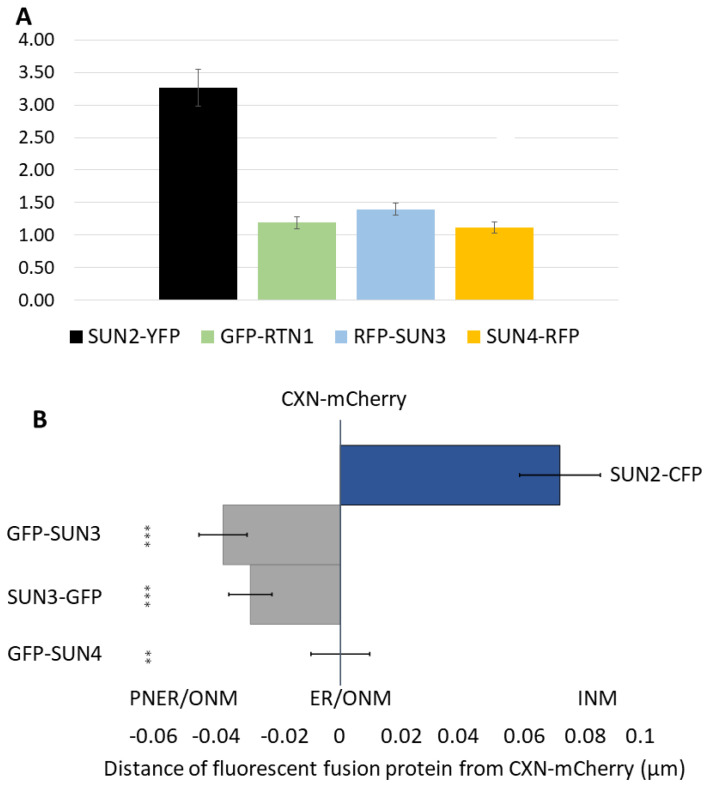
NE/ER fluorescence intensity (FI) ratios of ER and NE proteins. (**A**) NE/ER FI as measured for NE marker SUN2-YFP, ER marker GFP-RTN1, as well as for RFP-SUN3 and SUN4-RFP, when transiently coexpressed with p19 in *N. benthamiana* leaf epidermal cells. There were 30 cells analysed for each construct tested. There were statistically significant differences between group means as determined by Kruskal–Wallis H test, χ^2^(3) = 48.95, *p* = 1.377 × 10^−10^. (**B**) Proximity distances (µm) of fluorescent fusion proteins were computed with respect to the ER/ONM marker protein CXN-mCherry used as a reference (distance = 0 µm). SUN2-CFP (0.067 ± 0.012 µm), GFP-SUN3 (−0.036 ± 0.007 µm), SUN3-GFP (−0.028 ± 0.007 µm), and GFP-SUN4: (0 ± 0.009 µm) were measured by collecting 20 nuclei from each construct in three biological replicates.

**Figure 2 plants-12-01787-f002:**
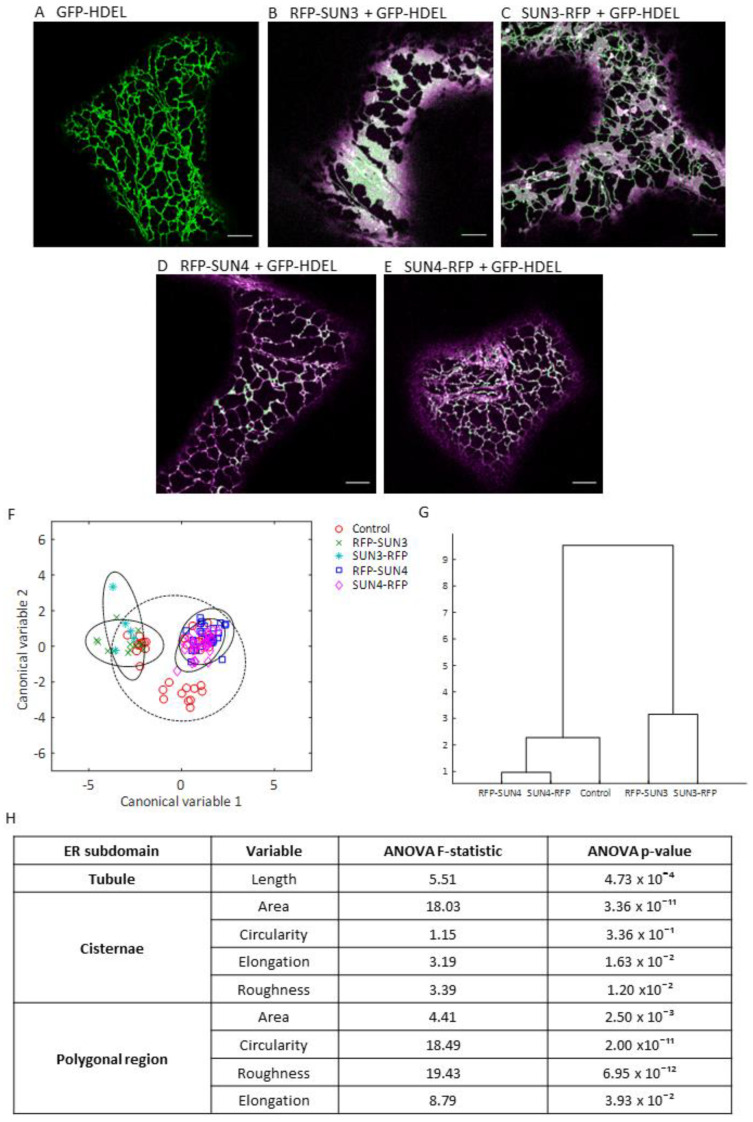
Overexpression of SUN3 and SUN4 affect ER structure. (**A**–**E**) Transient overexpression of GFP-HDEL (**A**) shown in green and (**B**) RFP-SUN3, (**C**) SUN3-RFP, (**D**) RFP-SUN4, and (**E**) SUN4-RFP shown in magenta. (**F**) Pairwise scatterplot of the first two canonical variables from the MANOVA analysis grouped by expressed proteins, the dotted line represents the 95% confidence limit. (**G**) Dendrogram of class means using MANOVA. (**H**) ANOVA results of planned comparisons of ER structure variables after ER structure was modified by SUN3 and SUN4 overexpression, compared to control ER structure. The F statistic and *p*-values (with applied Bonferroni correction) are shown. Images collected from five biological repeats, control, *n* = 35; RFP-SUN3, *n* = 13; SUN3-RFP, *n* = 5; RFP-SUN4, *n* = 26; SUN4-RFP, *n* = 27. Scale bars 5 µm.

**Figure 3 plants-12-01787-f003:**
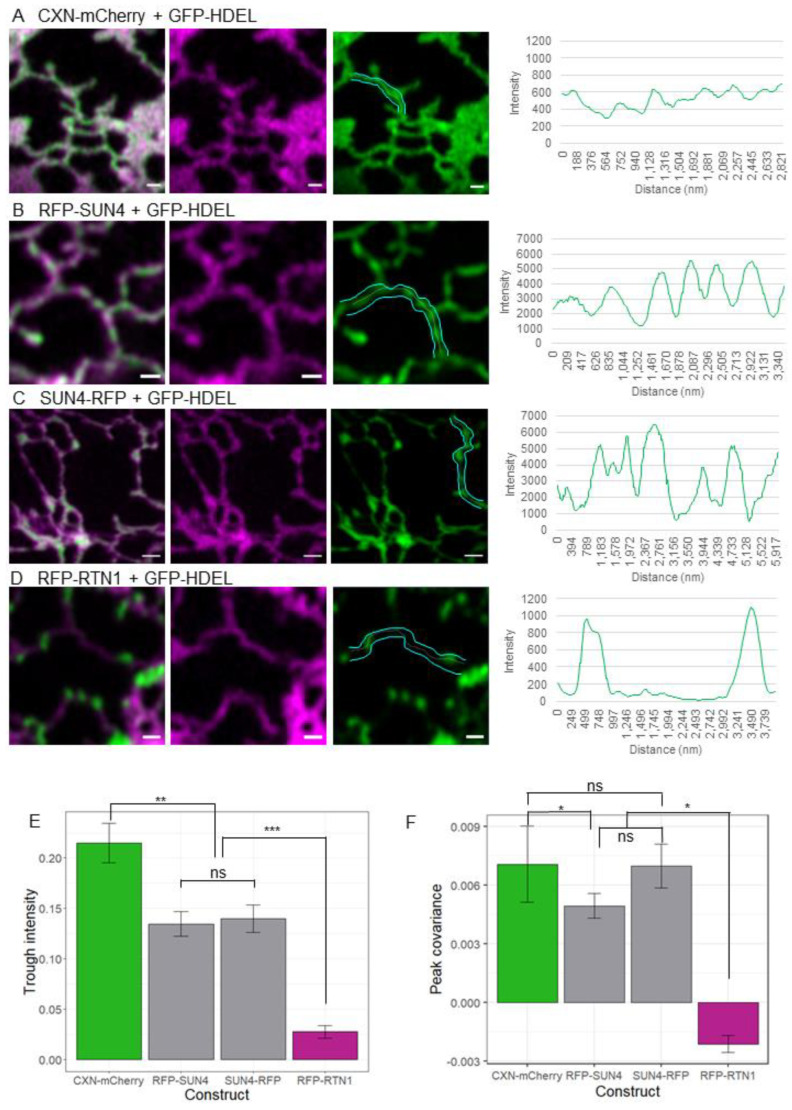
Comparison of ER constriction induced by SUN4 and RTN1 overexpression. High-resolution representative images of transmembrane proteins (magenta) and the ER luminal marker GFP-HDEL (green), alongside intensity traces along a single tubule, highlighted around the edge in cyan and along the centre of the tubule with a white dashed line. Images and line profiles shown for (**A**) CXN-mCherry, (**B**) RFP-SUN4, (**C**) SUN4-RFP, and (**D**) RFP-RTN1. Comparison of the mean trough intensity of the GFP-HDEL channel is shown (**E**), and the covariance of intensity peaks along all tubules in each image (**F**). Comparisons between groups using the Tukey HSD shown by asterisks, with *** representing *p*-value ≤ 0.001, ** representing *p*-value = 0.01–0.001, and * representing *p*-value = 0.05–0.01. Nonsignificant differences between the groups are denoted with ns. Results taken from three biological repeats with CXN-mCherry, *n* = 9; RFP-SUN4, *n* = 12; SUN4-RFP, *n* = 10; and RFP-RTN1, *n* = 9 cells analysed. Scale bars 2 µm.

**Figure 4 plants-12-01787-f004:**
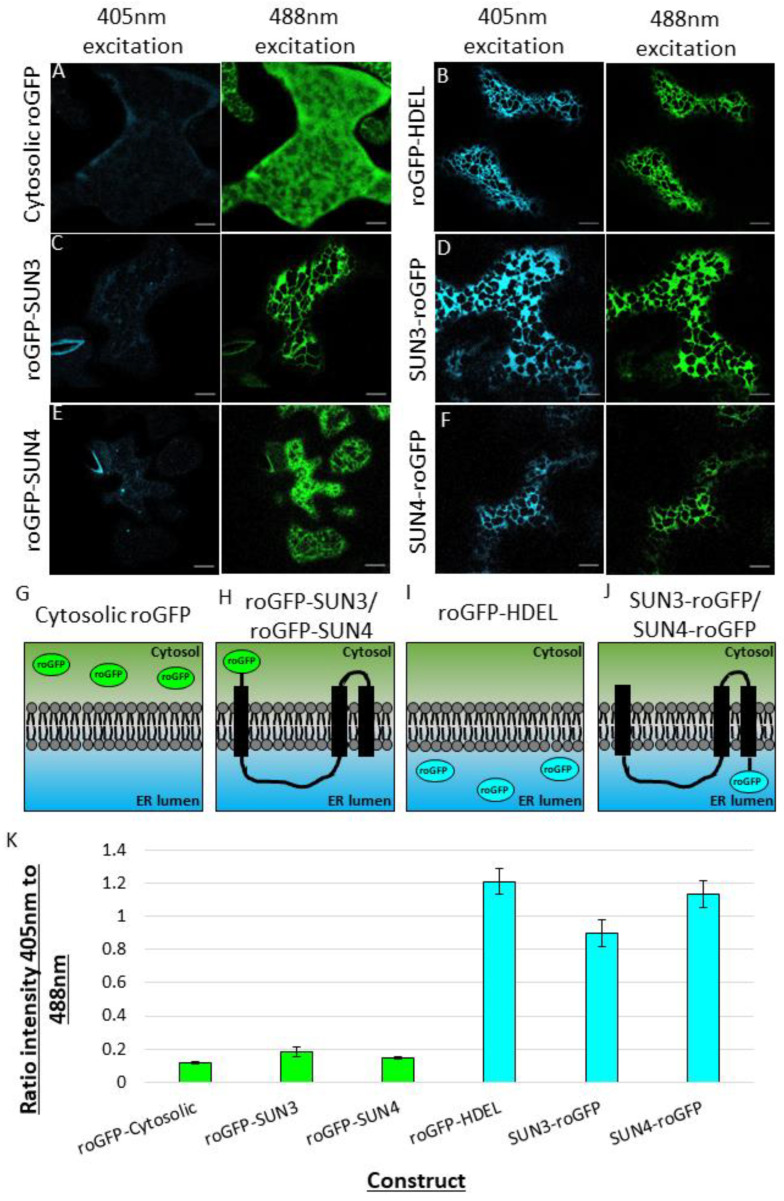
Characterising SUN3 and SUN4 membrane topology using a redox sensitive fluorescent protein (roGFP). Representative images of *N. benthamiana* epidermal cells transient expressing the redox sensitive fluorophore, roGFP, (**A**) localised to the cytosol, (**B**) the ER/NE lumen and attached to the N- and C-terminal of (**C**,**D**) SUN3 and (**E**,**F**) SUN4. All images collected in the emission range of 500–550 nm, with excitation by either a 405 nm laser (cyan) or a 488 nm laser (green). (**G–J**) Schematic representation of the membrane topology of SUN3 and SUN4. (**K**) A plot of the ratio of roGFP intensity after activation by 405 nm and 488 nm. Images collected across three biological repeats, (cytosolic roGFP, *n* = 10; roGFP-HDEL, *n* = 8; roGFP-SUN3, *n* = 12; SUN3-roGFP, *n* = 8; roGFP-SUN4, *n* = 9; SUN4-roGFP, *n* = 9). Scale bars 5 µm.

**Figure 5 plants-12-01787-f005:**
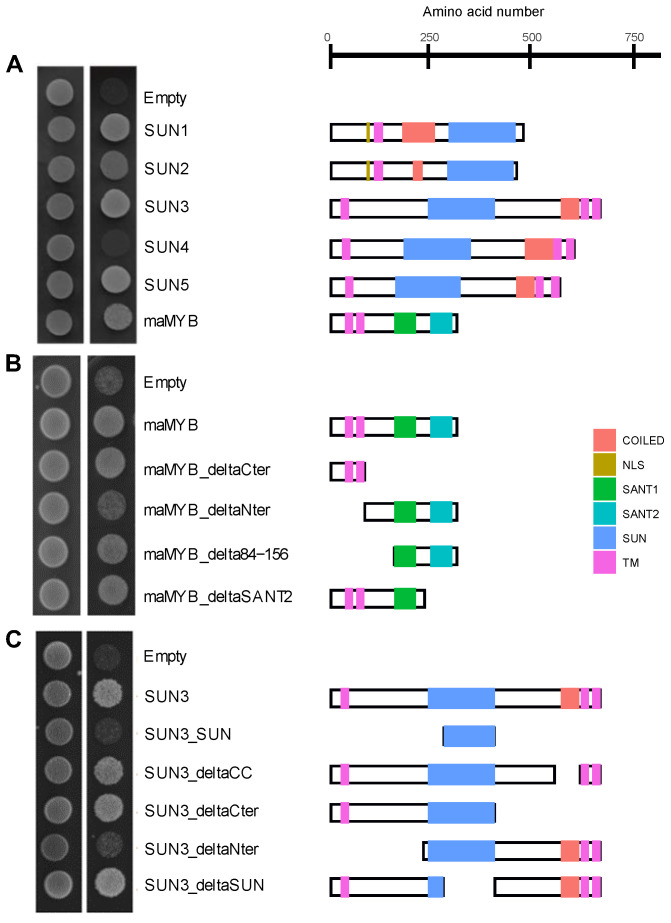
Interaction between maMYB and SUNs proteins in MYTH. All MYTH assays have been done on permissive and test media (PM and TM, respectively). Results from drop tests and schematic representation of protein structures drawn at scale are shown. (**A**) MYTH assays between full-length maMYB as bait and the SUN family proteins or maMYB as preys. Empty vector is used as a negative control. No interaction was observed with SUN4. (**B**) MYTH assays between full-length SUN3 as bait and deletions of maMYB as preys. Empty prey vector is used as a negative control. Weaker interaction was observed with maMYB-deltaNter. (**C**) MYTH assays between full-length maMYB as bait and deletions of SUN3 as preys**.** Empty prey vector is used as a negative control.

**Figure 6 plants-12-01787-f006:**
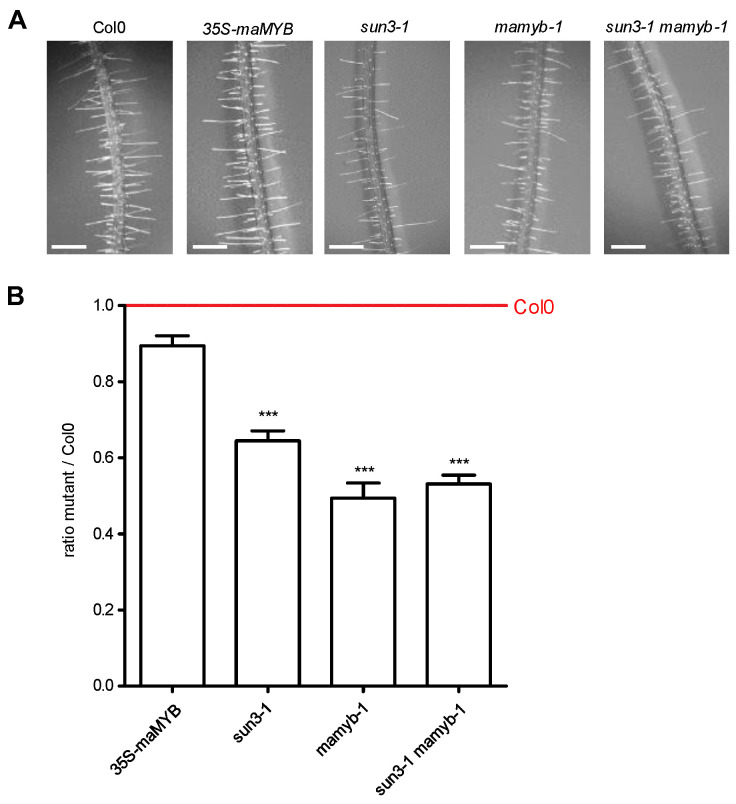
Mutant lines for sun3-1 and mamyb-1 show shorter root hair. (**A**) Representative images from root hairs for each genotype. Digital images were analysed with imageJ. At least 25 roots from seedlings were investigated and the five longest root hairs were measured at approximately 5 mm to the root tip, within the differentiation zone, for each plant as in Slabaugh et al. (2007). (**B**) Root hairs lengths are expressed as mutant/Col0 ratio (Col0 is set at 1, red line) for 35S-maMYB (*n* = 170), sun3-1 (*n* = 95), mamyb-1 (*n* = 40), and the corresponding double mutant line sun3-1 mamyb-1 (*n* = 85). Data are representative of five experimental replicates. Scale bar: 0.5 mm. *p* < 0.01 (***).

## Data Availability

All authors agree with MDPI Research Data Policies.

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
