# Peer review of "In Depth Topological Analysis of Arabidopsis Mid-SUN Proteins and Their Interaction with the Membrane-Bound Transcription Factor MaMYB"

_plants, 2023, doi:10.3390/plants12091787_

Round 1

Reviewer 1 Report

The paper investigates the mid-SUN family of conserved SUN proteins in plants. Previous studies have associated them with functions at the nuclear envelope and the ER. In this study, the authors used high-resolution confocal microscopy to characterize the localization of SUN3 and SUN4 in the perinuclear region, their topology, and their role in ER morphology. The study also found that SUN3 interacts with the ER membrane-bound transcription factor maMYB, both of which are involved in the regulation of root hair development, supporting the functional importance of mid-SUNs as an ER component. In my opinion, the manuscript is well-organized and clearly written, the experiments have been carefully designed and performed, and the results are appropriately interpreted. I have only a few minor suggestions and/or concerns to be addressed before its publication. 

1. I suggest adding representative fluorescence images (SUN2/3/4 colocalization with CXN at the peri-nuclear region) along with the quantification in Fig. 1B. In addition, Fig. 1A is missing a y-axis label and GFP-RTN1 color code.

2. While the use of the Kruskal-Wallis H test in Fig. 1A is appropriate for nonparametric ratio data, it is important to note that this test is an omnibus test statistic that only indicates if at least two groups are different, but cannot determine which specific groups are statistically significant. Similarly, the situation applies to the ANOVA test in Figure 2H. Therefore, I recommend removing the asterisks in Fig. 1B, which requires further posthoc tests. Alternatively, since the distance measurement is parametric, the authors could consider using t-tests for statistics.

3. I wonder if there might be a possibility that the anti-actin and anti-maMYB immunoblots in Supplementary Figure 2D were switched accidentally. It might be necessary to verify and clarify this point in the manuscript.

Author Response

We thank the reviewer for their positive comments and feedback. We have made changes accordingly - see details below - and feel this much improves the manuscript. 

Here our detailed response:

  1. I suggest adding representative fluorescence images (SUN2/3/4 colocalization with CXN at the peri-nuclear region) along with the quantification in Fig. 1B. In addition, Fig. 1A is missing a y-axis label and GFP-RTN1 color code.

Answer: A new supplementary figure S1 has been put together which includes representative images for the colocalisation of SUN2/3/4 markers with CXN. For figure 1, the colour code for the whole graph has been updated to make it easier to read. The y-axis are the sample labels.

  1. While the use of the Kruskal-Wallis H test in Fig. 1A is appropriate for nonparametric ratio data, it is important to note that this test is an omnibus test statistic that only indicates if at least two groups are different, but cannot determine which specific groups are statistically significant. Similarly, the situation applies to the ANOVA test in Figure 2H. Therefore, I recommend removing the asterisks in Fig. 1B, which requires further posthoc tests. Alternatively, since the distance measurement is parametric, the authors could consider using t-tests for statistics.

Answer: We indeed carried out a post-hoc test. We forgot to mention in the MS that following the Kruskal_Wallis test, we performed a Dunn’s test with Bonferroni corrections. This has now been added to the methods section.

  1. I wonder if there might be a possibility that the anti-actin and anti-maMYB immunoblots in Supplementary Figure 2D were switched accidentally. It might be necessary to verify and clarify this point in the manuscript.

Answer: We thank the reviewer for having noticed that the two Western blots were switched. Apologies for this - it has now has been corrected.

Reviewer 2 Report

In the manuscript of Andov et al., the authors showed a very interesting study that investigate in detail the localization, topology, and interaction of some mid-SUN proteins which are membrane specific. To do this, they used fluorescence microscopy tools, some over-expressed lines, tested the interaction with other proteins using the yeast two hybrids, and finally, they performed a genetic characterization using CRISPR-CAS. The approach to understanding the subcellular localization of the mid-SUN proteins (SUN3 and SUN4) was interesting because is simple but very direct. They found that SUN3 and SUN4 are located in the Endoplasmic reticulum (ER) and preferentially enriched in the outer nuclear membrane. Also, they found changes in the ER overexpressing SUN3 and SUN4 where the main differences are in tubule length, cisternae, and polygonal region areas. The topology of SUN3 and SUN4 was examined using the redox-sensitive fluorescent tag (two excitation peaks: 405 nm and 488nm) to locate each domain. The N-terminal of both SUN3 and SUN4 are located in the cytosol, the SUN domain, and coiled-coil domain are in the ER lumen, and finally, the two transmembrane domains are in the ER lumen. To provide some biological information about the function of the SUN3 at the ER,  they did a yeast-two hybrid screening to find SUN3-interactors. One strong interactor was detected, the transcription factor maMYB. 

The manuscript is a good piece of work that reveals the specific localization and topology of these mid-SUN proteins, but also the possible role of  SUN3 in the ER. The methodology and the experimental validations were properly used. This manuscript could be of potential interest to the journal. I have a few comments that could improve the manuscript.

Comments:

-The title is not attractive to the readers, maybe the authors could re-think about changing it.

-The figures have low quality. When zoomed in, the figures look blurry.

-Line 141: What do the authors think about this lack of correlation?

-Sometimes the phenotypes in single and double mutants are subtle. Is interesting the phenotype in the root hair length, however, I can see fewer root hairs also. Are the root hair number affected? Could the authors quantify this?

-From an evolutionary point of view, Which could be the advantage of having SUN3 and SUN4? Are SUN3 or maMYB present in plants without roots?

Author Response

We thank the reviewer for their positive comments and feedback. We have made changes accordingly - see details below - and feel this much improves the manuscript. 

Here our detailed response:

  1. -The title is not attractive to the readers, maybe the authors could re-think about changing it.

Answer: We have rewritten the title and hope this is now more attractive and to the point.

  1. -The figures have low quality. When zoomed in, the figures look blurry. Answer: The original figure files are of high resolution but the images were pasted straight into the text for the initial submission. Original figure files will be provided for the resubmission.
  2. -Line 141: What do the authors think about this lack of correlation? Answer: We comment on this lack of correlation in the sentence starting line 143 – we believe that the patchy appearance of lumenal GFP-HDEL is possibly due to membrane-localised SUN4 proteins having a mild constrictive effect hence while the SUN4 proteins appear evenly distributed in the membrane but the lumenal GFP-HDEL is restricted to patches. We have clarified this in the text by rewriting the sentence to: This suggests that membrane bound SUN4 was able to induce some ER lumen constriction resulting in GFP-HDEL patches - but not to the same extent and potential mechanism as RFP-RTN1 (Figure 3E).
  3. -Sometimes the phenotypes in single and double mutants are subtle. Is interesting the phenotype in the root hair length, however, I can see fewer root hairs also. Are the root hair number affected? Could the authors quantify this?

Answer: The number of root hairs per plant were evaluated at two positions: 3-9 and 9-12mm in respect to the root apex and we did not notice any significant variation among the tested genotypes. This is in good agreement with the original paper from Slabaugh, Held and Brandizzi 2011 describing that maMYB promotes root hair elongation but not root hair initiation. To clarify this, we changed the text as follows:

L257-261: As the dDisruption of maMYB was previously shown to affect root hair length but not root hair initiation using RNAi lines [21]. As previously described, we did not observe significant differences in root hair numbers among the tested genotypes. Then root hair elongation was investigated using a similar experimental procedure as described by Slabaugh et al. [21] (Figure 6A).

  1. -From an evolutionary point of view, Which could be the advantage of having SUN3 and SUN4? Answer: Are SUN3 or maMYB present in plants without roots? SUN3 is present in algae and moss while MaMYB is not (Supplementary Figure S1). Mosses are non-vascular plants, so they have a primitif root system called rhizoids. Evolutionary speaking, maMYB is only present in vascular plants, plants with a root system.

As proposed in the discussion l379-383, “Studies to date, including this, reveal interactions with Cter-SUNs and KASH proteins, potentially at the ONM [13], a subtle but distinctive role in ER morphology, interactions with a membrane transcription factor, the unfolded protein response pathway (Jaiswal et al., 2014), in pollen tube and root hair growth and as part of a chaperone complex guarding the ER sorting of LRR receptor [17].

It is also tempting to speculate that the ER localisation of midSUN could reflect the ancestral function and localisation of the SUN proteins before the emergence of the cell nucleus. During diversification midSUN could have acquired specific functions such as their interaction with maMYB and maybe its tranlocation to the nucleus.